# Surface Characteristics of Activated Carbon Sorbents Obtained from Biomass for Cleaning Oil-Contaminated Soils

**DOI:** 10.3390/molecules29163786

**Published:** 2024-08-10

**Authors:** Aitugan Sabitov, Meiram Atamanov, Ospan Doszhanov, Karina Saurykova, Kairat Tazhu, Almagul Kerimkulova, Adilkhan Orazbayev, Yerlan Doszhanov

**Affiliations:** 1Nanobiotechnology Laboratory, Combustion Problems Institute, Bogenbay Batyr Str., 172, Almaty 050040, Kazakhstan; aitugans@mail.ru (A.S.); amk1310@mail.ru (M.A.); saurykova.karina@mail.ru (K.S.); qairat.hairat@mail.ru (K.T.); almusha_84@mail.ru (A.K.); 2Faculty of Chemistry and Chemical Technology, Al-Farabi Kazakh National University, Al-Farabi Ave., 71, Almaty 050040, Kazakhstan; 3Faculty of Physics and Technology, Al-Farabi Kazakh National University, Al-Farabi Ave., 71, Almaty 050040, Kazakhstan; ospan.doszhanov@mail.ru; 4Faculty of Engineering and Information Technology, Almaty Technological University, Tole bi Str., 100, Almaty 050012, Kazakhstan; 5Faculty of Geography and Environmental Sciences, Al-Farabi Kazakh National University, Al-Farabi Ave., 71, Almaty 050040, Kazakhstan; orazbaevadilkhan75@gmail.com

**Keywords:** plant raw materials, activated carbon, biomass, adsorption activity, oil-contaminated soil

## Abstract

This study explores the sorption capacity and field application of activated carbons (ACs) derived from plant residues for the remediation of oil-contaminated soils. ACs were prepared from rice husks, reed stalks, pine sawdust and wheat straw using two-stage pyrolysis and chemical activation with potassium hydroxide. The structural and physicochemical properties of these ACs were analyzed using BET surface area measurements, SEM analysis, Raman spectroscopy and FTIR spectroscopy. Sorption experiments at room temperature demonstrated that AC from rice husks (OSL) exhibited the highest sorption capacities for gasoline, kerosene and diesel fuel, with values of 9.3 g/g, 9.0 g/g and 10.1 g/g, respectively. These results are attributed to the well-developed microporous and mesoporous structures of OSL, as confirmed by SEM images and a BET surface area of 2790 m^2^/g. Field tests conducted at the “Zhanatalap” oil deposit showed that the ACs effectively reduced the oil content in contaminated soils from 79.2 g/kg to as low as 2.6 g/kg, achieving a purification degree of up to 67% within 16 days. This study highlights the critical role of structural properties, such as porosity and graphitization degree, in enhancing the sorption efficiency of ACs.

## 1. Introduction

Activated carbon (AC) is a highly versatile and widely used nanostructured material employed to purify and remove pollutants from the atmosphere, water and soil, increase the specific capacity in energy-storage systems and support numerous other applications [1,2,3,4]. The advantages of AC-based sorbents over other materials (such as zeolites, clay minerals, silica gels, etc.) are attributed to their significantly larger specific surface area, which can reach up to 3000 m^2^/g, their well-developed porous structure, chemical stability and the ability to control surface chemistry by modifying the number and types of functional groups.

The increased interest in research on these sorbents is driven by the continuous improvements in chemical-thermal activation methods and the identification of the most promising precursors [5,6]. One of the renewable and cost-effective precursors is plant waste, which is available in large volumes, locally abundant and often burned, leading to environmental disadvantages and inefficient use [7]. Recently, many AC precursors have been studied, including acorn shells [8,9], pistachio nut shells [10], grape seeds [11], corn cobs [12], banana stems [13], apricot kernels [14], sawdust [15,16], rice husks [17,18], walnut shells [19], sugarcane bagasse and sugar beet pulp [20].

AC is produced using two main methods: physical and chemical activation. Physical activation involves the carbonization of the raw material followed by treatment at high temperatures in the presence of active agents such as water vapor or carbon dioxide [21,22]. This method results in AC with high porosity and a large surface area. Chemical activation begins with the impregnation of the raw material with chemicals such as KOH, H_3_PO_4_, ZnCl_2_ or H_2_SO_4_, followed by heat treatment, which promotes the formation of a porous structure in the carbon. Recently, increasing attention has been paid to physicochemical activation, which combines elements of both methods. This approach enhances the properties of AC, increases its efficiency and broadens its range of applications.

Compared to chemical activation, physical activation occurs at higher temperatures typically ranging from 700 °C to 1100 °C [23]. The dehydrating effect of the active substances used in chemical activation inhibits tar formation [24,25], leading to higher yields of porous carbon and lower activation temperatures compared to physical activation methods [26,27]. Prahas et al. reported that chemical activation methods, using agents such as H_3_PO_4_ and ZnCl_2_, are effective for activating previously carbonized lignocellulosic materials. On the other hand, agents like KOH are suitable for activating charcoal or charcoal-like precursors. Combining physical activation with air makes heat treatment cheaper due to the reduced energy consumption and shorter activation times [28,29,30,31].

Rice husks from the genus Oryza (OSL), reed stalks from the genus Phragmites (PhAC), pine sawdust from the genus Pinus (PSL) and wheat straw from the genus Triticum (TAL) are abundant plant species native to Central Asia and Kazakhstan. Due to their availability and low cost, these residues are well-suited as raw materials for producing AC. However, to our knowledge, despite their potential applications, there have been no reported studies on obtaining OSL, PhAC, PSL and TAL using a two-stage KOH chemical activation process under air.

The primary objective of this study was to produce AC from biomass plant residues OSL, PhAC, PSL and TAL using KOH as the activating agent in a two-stage process designed to minimize energy consumption at lower air temperatures. Pyrolysis was conducted at various activation temperatures ranging from 300 °C to 850 °C, with a heating rate of 5 °C/min. The study investigated the influence of activation temperature and time, as well as the KOH-to-carbon ratio, on the AC derived from biomass. The optimal activation conditions for each biomass type OSL, PhAC, PSL, TAL were determined by analyzing parameters such as iodine number and specific surface area of the prepared AC.

Incorporating AC into soil serves as a long-term carbon-storage reservoir, delaying its release into the atmosphere as CO_2_, which contributes positively to climate change mitigation efforts. An additional advantage is that the raw materials for AC production originate from renewable sources and organic waste from the agro-industry, which often pose environmental pollution challenges [32,33,34,35].

In the Republic of Kazakhstan, soil contamination with oil poses a significant environmental challenge, where AC has shown promising applications. Its main competitive advantages include high sorption capacity for petroleum products, effective degradation of oil and petroleum substances, cost-effectiveness of reclamation technology and rapid reclamation process. Furthermore, the production of modified AC is environmentally less harmful.

This study investigates the physico-chemical properties of AC derived from various plant residue sources in Kazakhstan, including wheat straw, rice husks, reed stalks and pine sawdust. Additionally, it explores the application of these AC samples to enhance soil fertility and remediate oil-contaminated areas near the “Zhanatalap” deposit in the Atyrau region.

## 2. Results

### 2.1. Physico-Chemical Characteristics of Activated Carbon Samples

The main physico-chemical characteristics of the different AC samples are presented in Table 1. The bulk densities for the carbonized samples of OSL, PhAC, PSL and TAL were 0.482 kg/m^3^, 0.396 kg/m^3^, 0.423 kg/m^3^ and 0.388 kg/m^3^, respectively. Crushed OSL consisted of fine-grained particles sized 0.25–0.35 mm and highly dispersed particles smaller than 40 microns. PSL particles were commensurate, with an average size range of 0.15–0.25 mm. TAL and PhAC were easily crushed and contained carbon dust, with average particle sizes of 0.1–0.2 mm.

The nitrogen adsorption-desorption isotherm at low temperature (−196 °C) was used to analyze the physical properties of the AC. The BET surface areas for OSL, PhAC, PSL and TAL were 2790, 1330, 2305 and 1755 m^2^/g, respectively.

The pore structure analysis of the AC samples reveals significant differences in their micropore volume, pore size and distribution. Table 2 presents detailed pore volume measurements using the Dubinin-Radushkevich (D-R), Barrett-Joyner-Halenda (BJH) and Dubinin-Astakhov (D-A) methods.

By the D-R equation, the micropore volume was highest for OSL at 0.712 cm^3^/g, followed by PSL at 0.665 cm^3^/g, TAL at 0.576 cm^3^/g and PhAC at 0.490 cm^3^/g. The BJH method provided additional insights into the mesopore volume. The best-activated sorbent, OSL, had a mesopore volume of 1.46 cm^3^/g, followed by PSL at 1.19 cm^3^/g, TAL at 0.88 cm^3^/g and PhAC at 0.43 cm^3^/g. Further analysis using the D-A method highlighted the pore size distribution where OSL again showed the highest volume at 0.84 cm^3^/g, followed by PSL at 0.74 cm^3^/g, TAL at 0.65 cm^3^/g and PhAC at 0.59 cm^3^/g. The results indicate that the AC have a well-developed meso- and microporous structure with a substantial volume of micropores. 

Comparing the results from Table 1 and Table 2, OSL exhibits the most favorable characteristics for adsorption applications. OSL shows the highest BET surface area, bulk density and significant micropore and mesopore volumes. The data from Table 2 further supports this, with OSL having the highest D-R, BJH and D-A volumes. PSL also shows high BET surface area and substantial pore volumes, making it another effective adsorbent. In contrast, PhAC and TAL have lower surface areas and pore volumes, indicating less developed porous structures but still viable for specific applications. Overall, OSL and PSL perform the best among the samples, highlighting the importance of selecting the right biomass and activation conditions to achieve optimal adsorption performance.

### 2.2. Structural Characteristics of Activated Carbon Samples

Figure 1a–d shows SEM images of the morphology of AC samples from different plant sources. The images highlight distinct differences in the surface structures and porosity of the samples. Figure 1a (OSL) reveals a rough surface with a high density of irregular, undisturbed particles. The presence of numerous cracks and crevices suggests a well-developed porous structure [36]. Figure 1b (PhAC) displays a similar porous morphology, with extensive film-like structures and large voids, indicative of efficient activation and pore formation.

In contrast, Figure 1c (PSL) shows a more uniform surface with fewer visible pores and less evidence of structural disruption. The circular patterns and smooth areas suggest limited activation, correlating with lower micro- and mesopore volumes [37]. Similarly, Figure 1d (TAL) presents a relatively smooth and undisturbed surface, with fewer cracks and crevices compared to OSL and PhAC, indicating a less developed porous structure.

These observations are consistent with BET analysis, which confirms the presence of macropores, mesopores and micropores in OSL and PhAC, while PSL and TAL exhibit smaller pore volumes. The SEM analysis underscores the significant impact of activation methods on the morphological characteristics of AC derived from different biomass sources.

### 2.3. Optimization of Activation Parameters for Enhancing the Adsorption Capacity of Biomass-Derived Activated Carbons

The observed trends in Figure 2a–d indicate the influence of activation parameters on the iodine number of AC derived from various biomass sources. Figure 2a shows that the iodine number increases with the S/L ratio up to 1:3 (up to 2700 mg/g) and then decreases at a 1:4 ratio (around 2450 mg/g). This trend suggests that an optimal concentration of the activation agent is crucial for maximizing the iodine number. Beyond this optimal ratio, excess KOH can lead to pore collapse and blockage, reducing the effective surface area for adsorption [38].

Excess KOH can adversely impact the pore structure of AC due to over-activation, pore blockage and destruction of pore walls. High concentrations of KOH can lead to over-activation, causing micropores to collapse and merge into larger, less effective pores, thereby reducing the overall surface area available for adsorption [39]. The aggressive nature of KOH at high concentrations can also destroy the carbon pore walls, leading to the merging of adjacent pores and significantly reducing the material’s adsorption capacity [40]. Therefore, maintaining an optimal impregnation ratio is crucial to balance the development of the porous structure and the preservation of the carbon framework, as exceeding this ratio results in structural damage that outweighs the benefits of increased activation.

Figure 2b presents the effect of the impregnation coefficient for samples preheated at 250 °C and then activated at 800 °C. Similar behavior is observed, where the iodine number increases (up to 2800 mg/g) with the S/L ratio up to 1:3 and then decreases at 1:4 (around 2600 mg/g). This confirms that an optimal balance of the activation agent enhances the porous structure while excessive amounts can be detrimental [41].

Figure 2a,b show a similar trend in the relationship between the S/L ratio and the iodine number, with the highest iodine number achieved at an optimal S/L ratio of 1:3. However, preheating the samples at 250 °C before activation, as shown in Figure 2b, can influence the development of adsorption characteristics. Preheating helps in the initial decomposition of volatile components, which can aid in the formation of a more stable and developed porous structure during the subsequent high-temperature activation [42].

Also, the thermal treatment helps in the partial carbonization of the precursor material, leading to the stabilization of the carbon structure. This stabilization is crucial for the formation of a more developed and stable porous network during subsequent high-temperature activation. For instance, preheating lignocellulosic materials such as rice husks before activation can lead to a higher surface area and pore volume, resulting in better adsorption performance [43].

Figure 2c illustrates the effect of activation temperature on the iodine number of AC derived from various biomass sources. The iodine number increases with temperature up to around 750 °C and then decreases at 800 °C. This trend suggests that excessively high temperatures may cause structural damage or excessive burn-off, reducing the adsorption capacity [44]. The optimal activation temperature for maximizing the iodine number appears to be around 750 °C, beyond which the structural integrity of the carbon material is compromised due to those mentioned above reducing the effective surface area.

OSL and PSL show substantial increases in iodine number up to 750 °C, indicating good thermal stability and effective porous structure development. In contrast, PhAC and TAL exhibit more modest increases, suggesting lower thermal tolerance and less effective activation. OSL and PSL, with higher specific surface areas, demonstrate superior adsorption characteristics at optimal temperatures, while PhAC and TAL, with lower specific surface areas, show reduced adsorption capacities. These variations highlight the importance of selecting appropriate biomass materials and optimizing activation conditions to achieve high-performance AC.

Figure 2d shows the effect of activation time, with iodine numbers peaking at around 3 h and then declining. Prolonged activation times may lead to similar issues as excessive activation agent ratios and temperatures, such as pore collapse or blockage. OSL shows significantly higher iodine numbers up to 2800 mg/g than the other precursors, indicating superior performance and more effective activation within the optimal 3 h timeframe. PSL also demonstrates substantial increases in iodine number more than 2540 mg/g, but to a lesser extent than OSL. In contrast, PhAC and TAL exhibit less pronounced increases and earlier declines in iodine number around 2430 and 2250 mg/g, respectively, indicating lower tolerance to extended activation times. OSL, with its higher specific surface area and developed porosity (Table 2 and Figure 1), performs exceptionally well under optimal activation times, while PhAC and TAL, with lower specific surface areas, are more susceptible to structural damage from prolonged activation. 

Similar results have been reported in the literature, where different biomass sources were used to produce AC and the effects of activation parameters were studied [45,46,47]. Research has shown that optimal activation time is crucial for maximizing adsorption capacity, with prolonged times leading to decreased performance due to pore collapse and blockage. Studies have observed that the iodine number and surface area increase with activation time up to an optimal point, beyond which further activation causes structural damage and reduces adsorption capacity [48,49]. Additionally, the importance of carefully controlling activation time and temperature to develop a stable and efficient porous structure has been emphasized, aligning with our findings on the need for optimal activation parameters [50].

### 2.4. Characterization of Biomass-Derived Activated Carbons: Insights from FTIR and Raman Spectroscopy

The FTIR spectra presented in Figure 3a of the AC samples OSL, PhAC, PSL and TAL reveal several common functional groups typically found in biomass-derived AC. Key peaks in the spectra include those for hydroxyl groups (3430 cm^−1^), aliphatic C-H stretching (3120 cm^−1^), carbonyl groups (1740 cm^−1^), aromatic C=C stretching (1640 cm^−1^) and C-O stretching in ethers and alcohols (1000–1300 cm^−1^) [51,52,53]. These peaks are essential for understanding the chemical transformations that occur during the activation process.

A notable feature across all samples is the broad hydroxyl peak, indicating moisture retention and the presence of phenolic compounds, which are common in biomass. The peak at 3120 cm^−1^ shows significant variation among the samples, with PhAC exhibiting the highest intensity, suggesting a greater presence of aliphatic C-H groups [54]. This implies a lower degree of graphitization, with PhAC retaining more of its original aliphatic structure compared to OSL and PSL, where this peak is less intense. The reduced aliphatic content in OSL and PSL suggests higher graphitization and more extensive aromatic condensation, which can enhance their ability to adsorb non-polar compounds such as oil due to increased surface hydrophobicity and π–π interactions [55].

The carbonyl peak at 1740 cm^−1^ is distinct in all samples, indicating varying levels of oxidation, which can influence the chemical reactivity and adsorption properties of the AC. Although the aromatic C=C stretching peak at 1640 cm^−1^ is partially obscured by the adjacent carbonyl peak, its presence across the samples suggests the formation of stable aromatic systems essential for adsorption processes involving π–π interactions [56]. These differences in the FTIR spectra reflect how the degree of functionalization and graphitization can tailor the AC for specific applications, such as adsorbing oil spills, emphasizing the need to balance these properties to optimize performance in environmental remediation.

The Raman spectra analysis depicted in Figure 3b for the AC samples—OSL, PhAC, PSL and TAL—reveals critical insights into their structural properties and degrees of graphitization. Using Lorentzian curve deconvolution, the analysis isolates key Raman modes D*, D, D″, G and D′ bands, allowing for precise differentiation between graphitic and disordered components. This method surpasses the traditional I(D)/I(G) ratio by overcoming inaccuracies from band overlap, providing clearer insights into each sample’s structural features. The degree of graphitization (*Gf*) was quantified using the equation [57,58]:$$Gf=\frac{A\left(G\right)}{A_{total}}$$
where A(G) is the area under the G peak, and *A_total_* is the total area under all peaks. TAL exhibited the highest graphitization at 69%, suggesting a predominantly graphitic structure with enhanced mechanical strength and thermal stability. This makes TAL ideal for applications demanding high durability and structural integrity. Conversely, OSL displayed the lowest graphitization at 48%, indicating a significant presence of disordered or amorphous carbon structures, which enhances its adsorption capabilities due to more active sites and a larger surface area. PhAC and PSL, with graphitization degrees of 58% and 52%, respectively, demonstrate a balance between crystalline and amorphous phases, potentially offering versatile chemical and physical properties beneficial for medium-duty adsorption tasks. The differences in graphitization levels have a direct impact on the adsorption properties of these AC. Higher graphitization, as seen in TAL, typically leads to fewer surface functional groups, reducing adsorption sites. In contrast, lower graphitization retains more functional groups, increasing adsorption sites for various adsorbates and enhancing overall adsorption capacity.

### 2.5. Investigation and Field Application of Activated Carbon for Hydrocarbon Sorption

The potential use of AC samples for the remediation of contaminated soils was investigated through sorption experiments. The sorption capacities of AC for gasoline, kerosene and diesel fuel were evaluated and detailed in Table 3. Among the samples, OSL showed the highest sorption capacities: 9.3 g/g for gasoline, 9.0 g/g for kerosene and 10.1 g/g for diesel fuel. These excellent results are attributed to the well-developed microporous and mesoporous structures of OSL, as evidenced by the nitrogen adsorption/desorption isotherms. In contrast, PhAC showed moderate sorption capacities, averaging 7.0 g/g for petroleum products, while PSL and TAL showed lower capacities, ranging from 4.5 to 5.1 g/g. The higher sorption of diesel fuel compared to gasoline and kerosene in all samples may be due to the higher viscosity of diesel fuel, which enhances its interaction with carbon surfaces.

To confirm the results in real conditions, field tests were conducted at the Zhanatalap oil field in the Atyrau region (Figure 4). Two sites were designated: a control site and a site treated with AC obtained from rice husk, reed stalks, pine sawdust and wheat straw. AC was introduced into the oil-contaminated soil, and samples were collected periodically from 8 to 16 days after treatment to estimate the oil content using the gravimetric method. Figure 5a shows the initial and subsequent oil content in the soil samples.

The initial soil contamination was 79.2 g/kg. After 16 days of treatment, the oil content in the soils with the addition of AC decreased to 2.6–3.3%, indicating a remediation rate of 58.2–67.1%, as shown in Table 4. The field results are consistent with the laboratory data, demonstrating that these AC are effective in the sorption remediation processes of oil spills. The study highlights the potential of using natural plant-based AC for effective and environmentally friendly soil remediation. The structural and physicochemical characteristics of the AC samples play a decisive role in their sorption capabilities. Well-developed microporous and mesoporous structures in OSL, confirmed by surface area and pore volume measurements using the BET method, contribute to its high sorption capacity. These characteristics promote enhanced interactions with hydrocarbons, especially non-polar molecules, due to the increased surface area and the presence of π–π interactions, as suggested by Raman and FTIR analyses (Figure 3a,b). In comparison, the moderate graphitization and porous structures of PhAC and PSL result in lower sorption capacities. These results are consistent with literature data, where higher degrees of graphitization correlate with a decrease in adsorption sites for polar molecules, but enhance the adsorption of non-polar molecules due to hydrophobic interactions. The field application results support the laboratory results, highlighting the practical applicability of these materials for soil remediation in oil-contaminated environments.

At the outset of the experiments, two plots were designated: the first served as a control for oil-contaminated soil, while the second plot involved testing AC samples for remediation. Each plot measured 4 m^2^ (2.0 m × 2.0 m), with approximately 50 kg of soil per plot. AC derived from biomass plant residues (specifically rice husks, reed stalks, pine sawdust and wheat straw) activated with potassium hydroxide through two-stage pyrolysis in air was introduced into the oil-contaminated soil.

Following the application of AC, the soil was appropriately moistened and aerated. Soil samples were then periodically collected between 8 and 16 days after treatment to assess the oil content. The determination of oil content in soil samples was carried out using the weight method, involving extraction of hydrocarbons from soil samples using hot hexane in a Soxlet apparatus.

Figure 5a shows the values of the oil content in contaminated soils. The initial degree of soil contamination with petroleum hydrocarbons was 79.2 g/kg or 7.9 wt.%. After applying AC samples to oil-contaminated soil, the oil content gradually decreases. On the soil samples with the addition of AC on the 8th day, the oil content was 5.7–6.7%, on the 16th day—2.6–3.3%. According to the oil content in soil samples, the degree of soil purification was calculated, the value of which after 16 days was 58.2–67.1% in soils with sorbent (Table 4). The conducted tests have demonstrated that the AC samples exhibit a high efficacy in soil purification. Specifically, after 16 days, the degree of purification of oil-contaminated soil reached 67.71%. 

The field tests conducted on real oil-contaminated soils underscore the effectiveness of utilizing pyrolyzed plant-derived AC based on biomass plant residues. Figure 5b shows the values of the oil content in contaminated soils during field work on the territory of oil-producing enterprises of the Atyrau region for cleaning with a biosorbent at certain intervals (16 days). On soil samples with the introduction of sorbents without modification on the 16th day, the oil absorption content was 1.7–4.1 g/g. According to the oil content in soil samples, the degree of soil purification was calculated, the value of which after 16 days was 3.0–4.3 g/g in soils with modified sorbents.

Having pronounced sorption properties, the modified sorbent in the soil immobilizes hydrocarbons on its surface, preventing their spread to adjacent media, but at the same time slowing down their physico-chemical destruction. In this regard, the use of a hydrophobic sorbent for soil purification is possible provided it is extracted from the soil after adsorption of petroleum products and further processing at a specialized landfill.

## 3. Materials and Methods

### 3.1. Preparation of Materials

This study investigated the characteristics of AC derived from biomass plant residues: OSL, PhAC, PSL, TAL. The starting materials were manually selected, purified with water, dried at 100–105 °C for 48 h in an oven and then milled and sieved to obtain particles sized between 1 and 2 mm. The resulting material was stored in airtight containers for subsequent experiments.

### 3.2. Activated Carbon Production from Agricultural Wastes

The production of AC from plant biomass was conducted in two stages:

(I) Impregnation and Preheating: Approximately 5 g of conditioned plant waste were impregnated with 1M KOH solutions at different sorbent/alkali ratios: (1:1), (1:2), (1:3) and (1:4) by weight. Specifically, for the 1:1 ratio, 5 mL of 1M KOH solution was used; for the 1:2 ratio, 10 mL of 1M KOH solution; for the 1:3 ratio, 15 mL of 1M KOH solution; and for the 1:4 ratio, 20 mL of 1M KOH solution. The impregnation process involved preheating from 30 °C to 80 °C and maintaining this temperature for 3 h to ensure thorough penetration of KOH into the biomass. The impregnated sorbents were then placed in a vertical electric furnace with an air supply. 

(II) Pyrolysis: In the second stage, pyrolysis of the samples was carried out at temperatures ranging from 300 °C to 850 °C with a heating rate of 5 °C/min. After reaching the maximum temperature, the samples were held for 3 h and then allowed to cool naturally to room temperature. During the cooling period, a continuous nitrogen flow was maintained to prevent oxidation and ensure the integrity of the AC. The resulting product underwent thorough washing with 0.1 M hydrochloric acid followed by distilled water to remove alkali residues, with continuous monitoring of the pH value of the wash solution (optimal pH 6–7). After washing, the AC samples were dried at 100–105 °C.

### 3.3. Study of Physico-Chemical, Technological Properties of Sorbents

After obtaining the AC samples, their bulk density and porosity were determined. To analyze the chemical composition of the samples, the QUANTA 3D 200i device (FEI, Williamsburg, MI, USA) was used in combination with an energy dispersive X-ray spectrometer EDAX (Jeol, Tokyo, Japan) and a semiconductor detector with an energy resolution of 128 eV (polymer material, sensitive area—d = 0.3 mm).

The bulk density of AC samples (kg/m^3^) was determined by weighing a measuring cylinder filled with a substance according to [59].

The bulk density is calculated using the formula:(1)$$\rho_{b}=\frac{m}{V}=\frac{5\times 10^{3}}{V}$$
where *ρ_b_*—the bulk density of the AC sample, kg/m^3^; *m*—the mass of AC, kg; *V*—the volume of AC in the cylinder after pre-sealing, m^3^.

Depending on the bulk density, AC can be classified as follows: *ρ_b_* > 2000 kg/m^3^—very heavy; 2000 > *ρ_b_* > 1100 kg/m^3^—heavy; 1100 > *ρ_b_* > 600 kg/m^3^—medium; *ρ_b_* < 600 kg/m^3^—light.

### 3.4. Sample Characteristics

The iodine number is a widely used method for assessing the adsorption capacity of AC due to its simplicity and ability to rapidly evaluate sorbent quality. It quantifies the porosity of the sorbents by measuring the amount of iodine adsorbed per gram of carbon (expressed as mg/g). The method involves impregnating AC samples with an iodine solution under ambient conditions, followed by filtration. The iodine content in the filtrate is then determined through titration, typically using a standardized iodine solution of 0.100 ± 0.001 mol/L concentration [60].

This adsorption capacity, often expressed in mg/g (typical range 500–1200 mg/g), indicates the level of activation, reflecting the micropore content in the AC (0–20 Å or up to 2 nm) and corresponding to a surface area ranging from 900 m^2^/g to 1100 m^2^/g [61]. AC produced OSL, PhAC, PSL and TAL through chemical activation with KOH demonstrate potential for industrial-scale development based on this criterion.

As a complementary surface assessment method, nitrogen adsorption-desorption isotherms were performed at −196 °C using a “Sorbtometer-M” (Granat Company, Saint-Petersburg, Russia) surface area analyzer after pre-gasification of samples at 250 °C for 24 h. BET method was applied to analyze the adsorption branch of the isotherms to determine surface areas. Microporosity was evaluated using the t-method, which combines micropore surface area and external surface area, and the volume of micropores was calculated using the (D-R) equation in logarithmic form. The surface tension of nitrogen was calculated at 8.72 MJ/m^2^, with a molar volume of 34.7 cm^3^/mol. Mesopore volume was determined using the (B-J-H) method by analyzing the desorption branch of the isotherm. The size distribution of micropores was assessed using the (D-A) method based on the adsorption isotherm at low relative pressure [62].

The morphology of the AC samples (OSL, PhAC, PSL, TAL) was examined using a SEM (model JSM-6490 LA) (Jeol, Tokyo, Japan) SEM images were obtained in secondary electron mode at 20 kV acceleration voltage and 20 pA radiation current, with samples prepared in powdered form. Samples were mounted on a copper holder using conductive glue or adhesive tape. Infrared spectra of the AC samples were recorded using a Spectrum 65, Perkin&Elmer 1100 FTIR (Waltham, MA, USA) spectrometer over the range of 4000–400 cm^−1^, employing KBr pellets with a resolution of 1 cm^−1^. The pellet for infrared studies was prepared by mixing a given sample with KBr crystals and pressed into a pellet. Raman spectroscopy was performed on a confocal Raman microscope (Jasco, NRS-3100 at «National nanotechnology laboratory of open type» Almaty, Kazakhstan).

### 3.5. Research of the Sorption Capacity of Activated Carbon in Relation to Hydrocarbons

Sorption activity in relation to oil products was studied under static conditions from model solutions. The model solutions were prepared as follows: aqueous solutions of oil products were obtained by mixing oil products with distilled water, followed by settling and separation of two phases in a Soxlet apparatus; water-oil emulsions were prepared by mixing water and oil products using a mechanical stirrer at high speeds. After sorption on AC samples, the initial and current concentrations of oil products were determined fluorimetrically using the standard technique for “Fluorat-02”. Immediately before the studies, the active coals of the 2.5 + 0.5 mm fraction were dried at a temperature of 105–110 °C for 2 h.

## 4. Conclusions

This study successfully demonstrated the potential of AC derived from plant residues as effective agents for the remediation of oil-contaminated soils. The results showed that rice husk AC (OSL) exhibited excellent sorption capacity for hydrocarbons, which was attributed to its extensive microporous and mesoporous structures. This finding is consistent with both laboratory tests and field applications, where the use of OSL resulted in significant reductions in oil contamination levels. These results highlight the important role of structural properties such as porosity and degree of graphitization in enhancing the sorption efficiency of AC.

The knowledge gained from this study highlights the potential of using natural plant-derived AC as a sustainable and environmentally friendly solution for soil remediation. The study also highlights the importance of optimizing activation processes to tailor the properties of AC to specific environmental applications. Future research could further explore the long-term effectiveness and reusability of these materials, as well as their effectiveness in a variety of environmental conditions and against a wider range of pollutants. Such efforts would strengthen the case for the large-scale use of biomass-derived AC in environmental cleanup operations.

## Figures and Tables

**Figure 1 molecules-29-03786-f001:**
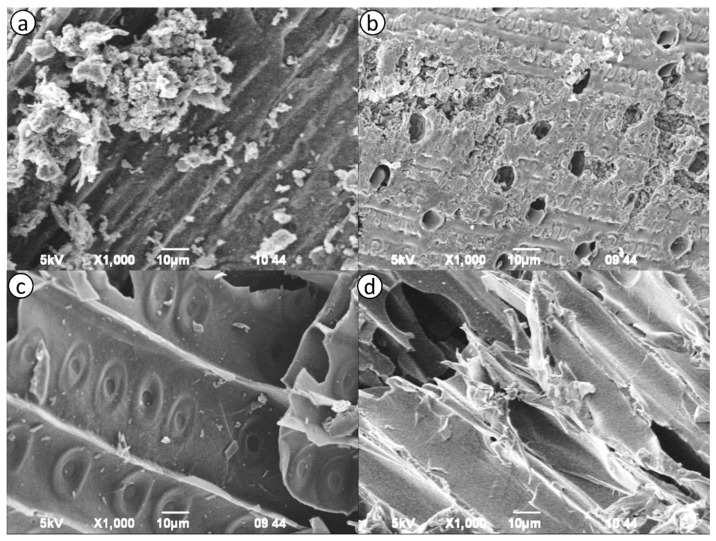
SEM images of the activated carbon structure: (**a**) OSL; (**b**) PSL; (**c**) PhAC; (**d**) TAL.

**Figure 2 molecules-29-03786-f002:**
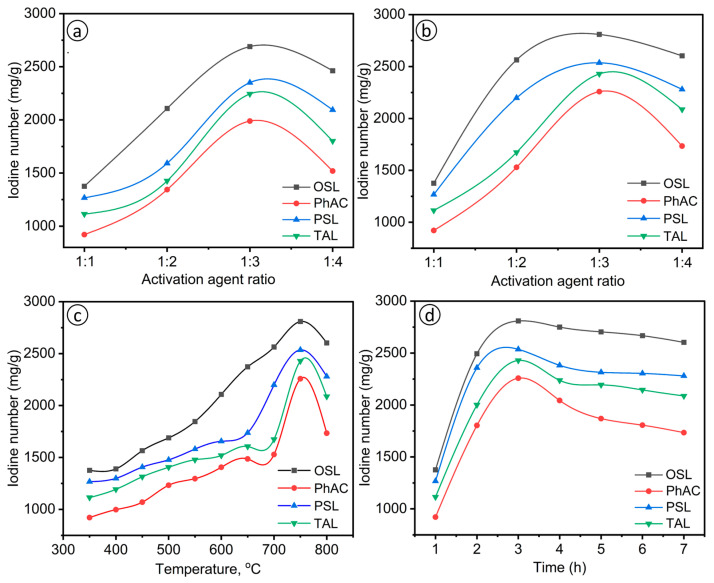
(**a**) Effect of the degree of impregnation at a preliminary activation temperature of 750 °C; (**b**) effect of the impregnation coefficient at an activation temperature of 250 °C; (**c**) effect of the activation temperature on the production of activated carbon; (**d**) effect of activation time on the production of activated carbon.

**Figure 3 molecules-29-03786-f003:**
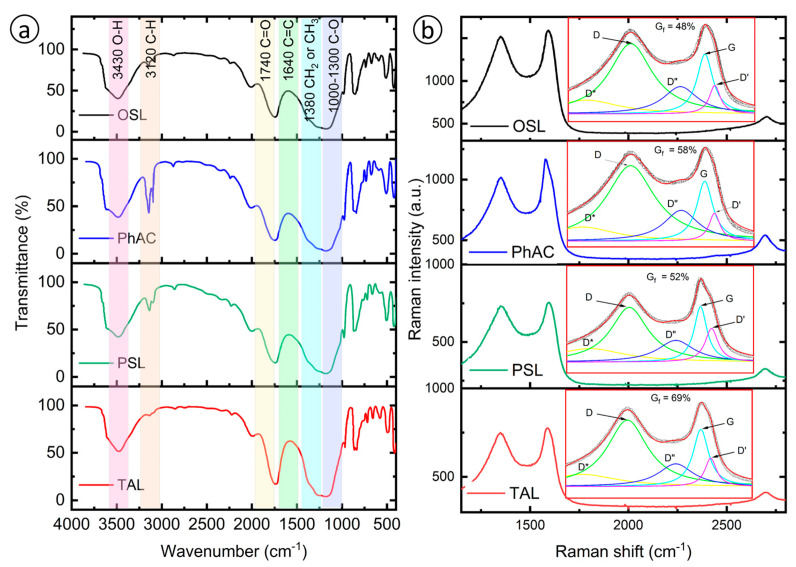
FT-IR (**a**) and Raman spectra (**b**) of the sorbent obtained taking into account the preliminary activation temperature (250 °C), activation temperature (750 °C), activation time (3 h) and the ratio of biomass-plant residues to KOH 1:3.

**Figure 4 molecules-29-03786-f004:**
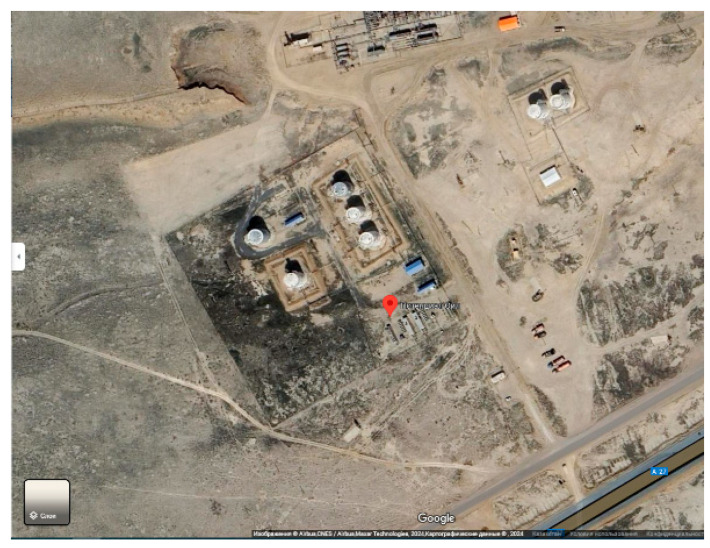
“Zhanatalap” deposit in the Isatai district of Atyrau region (a snapshot of the territory from Google Maps).

**Figure 5 molecules-29-03786-f005:**
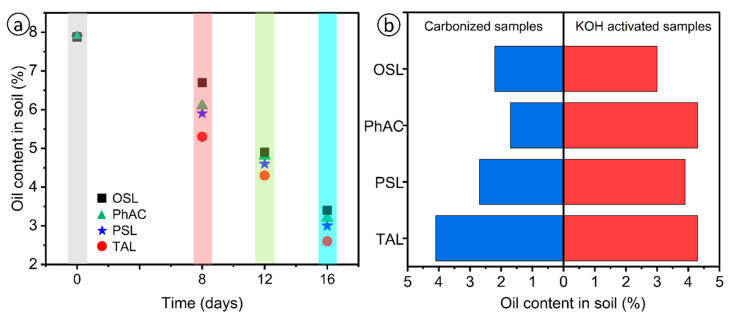
Properties of various materials (sorbents) for oil collection in the field (**a**) and oil content in soil depend on sorbent modification (**b**).

**Table 1 molecules-29-03786-t001:** Physico-chemical characteristics of activated carbon samples.

Activated Carbon	Bulk Density, kg/m^3^	Average Particle Size, mm	Specific Surface Area According to BET, m^2^/g	Volume of Micropores, cm^3^/g	Volume of Mesopores, cm^3^/g
OSL	0.482	0.25	2790	0.46	0.16
PhAC	0.396	0.15	1330	0.06	0.08
PSL	0.423	0.30	2305	0.26	0.09
TAL	0.388	0.20	1755	0.08	0.06

**Table 2 molecules-29-03786-t002:** Nitrogen adsorption isotherm and pore size distribution of activated carbon obtained taking into account the preliminary activation temperature (250 °C), activation temperature (750 °C), activation time (3 h) and the ratio of biomass-plant residues (OSL, PhAC, PSL and TAL) to KOH 1:3.

Activated Carbon	Volume of Pores According to (D-R), cm^3^/g	Volume of Pores According to (B-J-H), cm^3^/g	Volume of Pores According to (D-A), cm^3^/g	Relative Pressure, P/P^0^
OSL	0.665	1.19	0.74	0–0.5
PhAC	0.490	0.43	0.59	0–0.5
PSL	0.712	1.46	0.84	0–0.5
TAL	0.576	0.88	0.65	0–0.5

**Table 3 molecules-29-03786-t003:** Sorption capacities of activated carbons with respect to iodine and oil products.

Activated Carbon	Sorption Capacity		
	Gasoline, g/g	Kerosene, g/g	Diesel Fuel, g/g
OSL	9.30	9.00	10.10
PhAC	6.10	6.50	7.00
PSL	4.70	4.60	5.10
TAL	3.80	3.80	4.50

**Table 4 molecules-29-03786-t004:** Oil content in soil samples taken from an oil-contaminated site after treatment with activated carbons.

Number of Days	Activated Carbon	Degree of Soil Purification, %
0	-	0
8	OSL	27.8
	PhAC	25.3
	PSL	22.8
	TAL	15.2
16	OSL	67.1
	PhAC	60.8
	PSL	59.5
	TAL	58.2

## Data Availability

Data are contained within the article.

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
