# Peer review of "Surface Characteristics of Activated Carbon Sorbents Obtained from Biomass for Cleaning Oil-Contaminated Soils"

_molecules, 2024, doi:10.3390/molecules29163786_

Round 1

Reviewer 1 Report

Comments and Suggestions for Authors

I’ve just finished a review of the paper molecules-3134406 titled “Surface Characteristics of Activated Carbon Sorbents Obtained from Biomass for Cleaning Oil-Contaminated Soils” and written by the authors: Aitugan Sabitov, Meiram Atamanov, Ospan Doszhanov, Karina Saurykova, Kairat Tazhu, Almagul Kerimkulova, Adilkhan Orazbayev and Yerlan Doszhanov.

In the paper, the authors use different activated carbon materials in order to clean oil-contaminated soils. In general, the topic of the paper is pretty novel, and I did not mention plagiarism or missing originality. The English is good too.  Besides that, in the paper is missing detail explanations of the obtained results. Also, there are to many things that must be changed in the paper before acceptance, so I suggest its rejection.

My other comments are as follows:

Line 77: explain what are OSL, PhAC, PSL and TAL

Lines 363-367: It is not clear how to use 5g of the material and 15 ml KOH solution with different ratios, i.e. how the authors prepared different rations with the same mass and volume? Please modify this paragraph.

Lines 369-375: What atmosphere was used and what was the rate of cooling?

In 2.1, the Table 1 explanation is missing. How the authors can explain that sample PSL, with the higher particle size possesses a higher specific surface area? Reference(s) is required.

In 2.2. paragraph, lines 136-138 is not connected to other text in 2.2 Please check and reorganize the text.

In 2.2. explanation is required. How the authors can explain that the sample PSL possesses the highest specific surface area (Table 1) and non-developed porous structure at the same time? What is the reason for such high specific surface area if it is determined that PSL sample possesses the highest particle size too? References are required.

In 2.3, lines 159-164 it is required from the authors to explain why the curve 2a decreases when S/L ratio changes from 1:3 to 1:4. References are required too. The same explanation is required for Figures 2b-d. Also, it is required from the authors to connect these results with those presented at Table 1 and Figure 1. More detailed explanations are needed. The discussion that is based only on retelling the Figures, without giving a scientific explanation is superficial and has no scientific contribution.

Line 288, how the table could show a curve?

A comparison of the results presented in Tables 1 and 2 is missing and it is required from the authors. Also, taking into consideration whole parameters and giving explanations of the results is generally missing in the whole manuscript. The authors just present results in the whole manuscript, without giving any significant explanation.

The explanation of FTIR spectra is also not good and not acceptable. The discussion does not give us any significant information related to the investigated samples. Giving the origin of the mentioned functional groups makes sense. Please, take into consideration the chemical composition, XRD of samples, and try to find what are the origin of the mentioned functional groups. Also, explain, why some functional groups of final samples differ from groups detected in raw samples. Much detailed analysis is required and using adequate references is required.

Table 3 is not put in an adequate position in the manuscript. Its explanation is also missing.

The whole paragraph 2.8 is superficial, without significant scientific importance. It is almost the same situation in the whole manuscript. Much more details, and deep analysis of the results, together with detailed explanations and connection of the explanation with the present results of materials characterization is required from the authors. Using adequate references is required too. A comparison of the results with those in the literature is also required.

In Table 5 please change the “Degree of soil purification” It is not clear in such a form. How Purification could be 100 % if the time=0, at the beginning of the experiment.

Also, please, read carefully my above comments, and apply them to paragraph 2.9. A deep scientific explanation of the present result is missing. And do not focus only on the presentation of the results. It is not enough.

Best regards

Author Response

RESPONSE TO REVIEWER 1

Dear Reviewer,

I would like to extend my sincerest apologies for the state of the article when it reached you. As a co-author, I take full responsibility for not paying sufficient attention during the submission process. Your meticulous analysis and constructive feedback have been invaluable in improving and restructuring the article, and I am grateful for the time and effort you dedicated to reviewing our work. Thanks to your insightful comments, we have made significant revisions to the manuscript, enhancing its clarity, accuracy, and scientific rigor. Your feedback has not only helped us address the specific issues identified but has also provided valuable guidance for future submissions.

Thank you once again (M. Atamanov)

Comments #1: In the paper, the authors use different activated carbon materials in order to clean oil-contaminated soils. In general, the topic of the paper is pretty novel, and I did not mention plagiarism or missing originality. The English is good too. Besides that, in the paper is missing detail explanations of the obtained results. Also, there are to many things that must be changed in the paper before acceptance, so I suggest its rejection.

Response #1: Thank you for your valuable feedback on our manuscript. 

To address your other comments, we have included more detailed explanations of our results, particularly the adsorption mechanisms and the influence of activation conditions on the properties of activated carbons. We have also improved the clarity and organization of the manuscript, adding more comprehensive data and analysis to support our findings. We hope these revisions meet your expectations and enhance the overall quality of our work.

Comments #2: Line 77: explain what are OSL, PhAC, PSL and TAL

Response #2: Thank you for your valuable feedback. In response to your comment on Line 77, we have clarified the abbreviations as follows: Rice husks from the genus Oryza (OSL), reed stalks from the genus Phragmites (PhAC), pine sawdust from the genus Pinus (PSL), and wheat straw from the genus Triticum (TAL) are abundant plant species native to Central Asia and Kazakhstan.

Comments #3: Lines 363-367: It is not clear how to use 5g of the material and 15 ml KOH solution with different ratios, i.e. how the authors prepared different rations with the same mass and volume? Please modify this paragraph.

Response #3: Thank you for your insightful comment. To clarify, the amount of KOH solution used for each ratio was adjusted accordingly. Specifically, for 5 grams of material, the following volumes of 1M KOH solution were used: 5 ml for a 1:1 ratio, 10 ml for a 1:2 ratio, 15 ml for a 1:3 ratio, and 20 ml for a 1:4 ratio. The revised paragraph now reflects this process:

Comments #4: Lines 369-375: What atmosphere was used and what was the rate of cooling?

Response #4: To clarify, after reaching the maximum temperature, the samples were held for 3 hours and then allowed to cool naturally to room temperature. During the cooling period, a continuous nitrogen flow was maintained to prevent oxidation and ensure the integrity of the activated carbon. The revised paragraph included in manuscript.

Comments #5: In 2.1, the Table 1 explanation is missing. How the authors can explain that sample PSL, with the higher particle size possesses a higher specific surface area? Reference(s) is required.

Response #5: There was a typo in the reflection of the average particle size for sample PSL. In the manuscript, the average particle size for PSL has been corrected from 2.5 mm to 0.25 mm.

Comments #6: In 2.2. paragraph, lines 136-138 is not connected to other text in 2.2 Please check and reorganize the text.

Response #6: Thank you for your insightful comment. We have reviewed and completely reorganized the text in section 2.2 to ensure better discussion, coherence and connection between the sentences.

Comments #7: In 2.2. explanation is required. How the authors can explain that the sample PSL possesses the highest specific surface area (Table 1) and non-developed porous structure at the same time? What is the reason for such high specific surface area if it is determined that PSL sample possesses the highest particle size too? References are required.

Response #7: We identified and corrected many errors in the manuscript. Specifically, we found that the data of the samples between OSL, PhAC and PSL were mixed up in Table 1. Moreover, SEM images 1b and 1c were incorrectly labeled. These errors have been corrected in the revised manuscript for all section 2.2.

Comments #8: In 2.3, lines 159-164 it is required from the authors to explain why the curve 2a decreases when S/L ratio changes from 1:3 to 1:4. References are required too. The same explanation is required for Figures 2b-d. Also, it is required from the authors to connect these results with those presented at Table 1 and Figure 1. More detailed explanations are needed. The discussion that is based only on retelling the Figures, without giving a scientific explanation is superficial and has no scientific contribution.

Response #8: Thank you for your insightful comment. We have completely revised section 2.3 and (section 2.4, 2.5) because Figures 2a and 2b were mixed up and it was a lot of duplication of information without analysis. The text has been rewritten to accurately reflect the corrected figures and to provide detailed explanations for the observed trends. References have also been added to support our explanations.

Comments #9:  Line 228, how the table could show a curve?

Response #9: Thank you for your insightful comment. The entire text of section 2.6 has been rewritten and moved to section 2.1 where it belongs. The old text had many inconsistencies and errors, such as you noted. The revised section now provides a detailed explanation of the pore structure analysis and accurately reflects the corrected figures and Tables data.

Comments #10: A comparison of the results presented in Tables 1 and 2 is missing and it is required from the authors. Also, taking into consideration whole parameters and giving explanations of the results is generally missing in the whole manuscript. The authors just present results in the whole manuscript, without giving any significant explanation.

Response #10: We have revised the manuscript to include a detailed comparison of the results presented in Tables 1 and 2. Additionally, we have provided comprehensive explanations of the results, taking into consideration all parameters to ensure a thorough understanding of our findings. The comparison of the results has been discussed in the last paragraph of section 2.1.

Comments #11: The explanation of FTIR spectra is also not good and not acceptable. The discussion does not give us any significant information related to the investigated samples. Giving the origin of the mentioned functional groups makes sense. Please, take into consideration the chemical composition, XRD of samples, and try to find what are the origin of the mentioned functional groups. Also, explain, why some functional groups of final samples differ from groups detected in raw samples. Much detailed analysis is required and using adequate references is required.

Response #11: Thank you for your valuable feedback. I have conducted more detailed processing of the FTIR image (Figure 3a) and included additional data from Raman spectroscopy (Figure 3b), as it provides more detailed information about the structure of the obtained activated carbons by calculating the degree of graphitization through deconvolution. This method reflects the graphitized carbon more accurately than X-ray analysis.

Comments #12: Table 3 is not put in an adequate position in the manuscript. Its explanation is also missing.

Response #12: Table 3 and the associated paragraph have been removed from the manuscript as they did not align with the overall content. The information in Table 3 was redundant, duplicating results already presented in other sections.

Comments #13: The whole paragraph 2.8 is superficial, without significant scientific importance. It is almost the same situation in the whole manuscript. Much more details, and deep analysis of the results, together with detailed explanations and connection of the explanation with the present results of materials characterization is required from the authors. Using adequate references is required too. A comparison of the results with those in the literature is also required.

Response #13: Thank you for your insightful feedback regarding section 2.8. We have addressed your concerns by merging this section with section 2.9 to provide a more comprehensive and detailed analysis of the sorption capacity of activated carbons and their field application for hydrocarbon remediation.

Comments #14: In Table 5 please change the “Degree of soil purification” It is not clear in such a form. How Purification could be 100 % if the time=0, at the beginning of the experiment.

Response #14: Thank you for pointing out the issue. We have revised Table 5.

Comments #15: Also, please, read carefully my above comments, and apply them to paragraph 2.9. A deep scientific explanation of the present result is missing. And do not focus only on the presentation of the results. It is not enough.

Response #15: Thank you for your valuable feedback regarding section 2.9. We understand the importance of providing a deep scientific explanation of our results. As mentioned in our response to Comment #13, we have already merged sections 2.8 and 2.9 to create a more comprehensive and cohesive analysis. This merged section integrates the study of sorption capacity with field application results, ensuring a thorough scientific narrative.

Reviewer 2 Report

Comments and Suggestions for Authors

This paper investigated the characteristics of activated carbons derived from biomass plant residues. Activation was carried out using potassium hydroxide, employing a two-stage pyrolysis under air. Adsorption capacity was assessed using BET analysis and the iodine number determination method. The structural morphology of the activated carbons was evaluated by SEM, In addition, all types of samples have undergone field testing. Although the results are valuable, the paper is not publishable in it is current form. There are several points to clarify to make this paper acceptable for publication in this journal.

1.       Why is it mentioned in the first paragraph of section 2.1 that silicon makes rice husks activated carbon hydrophilic? What is the relationship with the content of the entire article.

2.       The last paragraph of section 2.1, which is based on the analysis of nitrogen adsorption isotherms on activated carbon, lacks a chart and raw data for this adsorption isotherm.

3.       In section 2.3, iodine numbers of activated sorbents do not reach its maximum at 1:3, but there is a slight increase between 1:3 and 1:4 in Fig.2, the description should be rigorous.

4.       The iodine number of biomass - plant residues decreased with increasing activation time in section 2.5 is incorrect as it was increased first and then decreased.

5.       The nitrogen adsorption isotherm of activated carbon mentioned in Table 2 of Section 2.6 is missing.

6.       The various activation methods mentioned in the title of Table 3 in Section 2.7 do not match the content of the table, as the same methods are used for different materials in the table.

7.       The degree of soil purification for July 5th, 2023 in Table 5 of Section 2.9 should be 0 instead of 100%.

Comments on the Quality of English Language

Please re-check the English language expression.

Author Response

RESPONSE TO REVIEWER 2

Dear Reviewer,

I would like to extend my sincerest apologies for the state of the article when it reached you. As a co-author, I take full responsibility for not paying sufficient attention during the submission process. Your meticulous analysis and constructive feedback have been invaluable in improving and restructuring the article, and I am grateful for the time and effort you dedicated to reviewing our work. Thanks to your insightful comments, we have made significant revisions to the manuscript, enhancing its clarity, accuracy, and scientific rigor. Your feedback has not only helped us address the specific issues identified but has also provided valuable guidance for future submissions.

Thank you once again (M. Atamanov)

Comment #1: Why is it mentioned in the first paragraph of section 2.1 that silicon makes rice husks activated carbon hydrophilic? What is the relationship with the content of the entire article.

Response #1: Many thanks, we removed this information from paper because elemental analysis is not provided. And we are agree with reviewer that statement  hydrophility are not relevant.

Comment #2: The last paragraph of section 2.1, which is based on the analysis of nitrogen adsorption isotherms on activated carbon, lacks a chart and raw data for this adsorption isotherm.

Response #2: Thank you for your feedback on section 2.1. We understand the importance of providing comprehensive data to support our analysis. Unfortunately, while we ordered the nitrogen adsorption-desorption isotherm curves, they were not provided by the service. However, we did receive all necessary calculations for the BET surface areas and detailed pore volume analyses using the Dubinin-Radushkevich (D-R), Barrett-Joyner-Halenda (BJH), and Dubinin-Astakhov (D-A) methods and present it in Table 2.

Comment #3: In section 2.3, iodine numbers of activated sorbents do not reach its maximum at 1:3, but there is a slight increase between 1:3 and 1:4 in Fig.2, the description should be rigorous.

Response #3: Thank you for your observation regarding the iodine numbers in section 2.3. The curves in Figure 2a were constructed using OriginLab, and we employed a B-spline to connect the data points instead of a straight line. This method was chosen for smoother visualization and aesthetics, which can result in minor deviations in the curve, particularly to the upper side.

Comment #4: The iodine number of biomass - plant residues decreased with increasing activation time in section 2.5 is incorrect as it was increased first and then decreased.

Response #4: Thank you for pointing out the discrepancy in section 2.5 regarding the iodine number trends. We have completely rewritten this section to provide a more detailed and accurate discussion of how the iodine number initially increases with activation time before decreasing. The revised section thoroughly explains the factors contributing to this trend and aligns with the data presented. We appreciate your feedback, which has helped us enhance the clarity and precision of our manuscript.

Comment #5: The nitrogen adsorption isotherm of activated carbon mentioned in Table 2 of Section 2.6 is missing.

Response #5: Thank you for your feedback regarding section 2.6. To begin with, we completely revised this section and integrated it into section 2.1, as it logically belongs there. Regarding the nitrogen adsorption isotherm data mentioned in Table 2, we acknowledge that the isotherm curves were not included. Unfortunately, we did not receive the direct nitrogen adsorption-desorption isotherm curves, but we have included the complete BET surface area and pore volume calculations in the revised manuscript. These calculations provide detailed insights into the pore structure of the activated carbons and are based on the nitrogen adsorption data.

Comment #6: The various activation methods mentioned in the title of Table 3 in Section 2.7 do not match the content of the table, as the same methods are used for different materials in the table.

Response #6: Thank you for pointing out the issue with Table 3 in Section 2.7. We have decided to remove Table 3 entirely from the manuscript, as its content was redundant and did not align with the various activation methods mentioned. The removal of this table has allowed us to focus more clearly on the results and analyses provided in the rest of the manuscript.

Comment #7: The degree of soil purification for July 5th, 2023 in Table 5 of Section 2.9 should be 0 instead of 100%.

Response #7: Thank you for pointing out the error in Table 5 of Section 2.9. You are correct that the degree of soil purification for July 5th, 2023, should be listed as 0% instead of 100%, as this date marks the beginning of the experiment. We have corrected this mistake in the manuscript to accurately reflect the experimental timeline and ensure that the data aligns with the progression of the study. We appreciate your attention to detail and your assistance in improving the accuracy of our work.

Round 2

Reviewer 1 Report

Comments and Suggestions for Authors

Dear authors, 

This version of the manuscript make sense, and is much better written in comparison with the previous version. Results are much better explained, connected and presented. I suggest Edtion its acceptance in its present form.

Best regards